# TVI-CoT: Text-Visual Interleaved Chain-of-Thought Reasoning for Multimodal Understanding

**Lianyu Hu**[1]  **Xiaoyu Ma**[1]  **Zeqin Liao**[1 ✉]  **Yang Liu**[1]

## Abstract

Chain-of-thought (CoT) reasoning has proven effective for enhancing problem-solving in large language models. However, when applied to multimodal LLMs (MLLMs), existing CoT approaches suffer from a fundamental limitation: *they perform reasoning entirely in text without accessing visual features during the reasoning process*. After initial visual encoding, image information becomes inaccessible, forcing models to reason based solely on whatever was captured in the initial description, which forms a "vision-blind reasoning" paradigm that limits fine-grained visual extraction, error verification, and adaptive attention. We propose Text-Visual Interleaved Chain-of-Thought (TVI-CoT), a framework that enables explicit interleaving of textual reasoning and visual feature access through learnable control tokens (⟨Think⟩, ⟨Look⟩, ⟨Answer⟩). These tokens allow dynamic switching between reasoning and visual grounding, attending to relevant image regions conditioned on the evolving reasoning state. Experiments on eight benchmarks demonstrate state-of-the-art results among MLLM-based CoT methods and notable performance boost compared to the baseline: +6.1% on MMMU, +3.8% on MathVerse, +3.4% on MathVista, and +3.4% on ScienceQA. Code is available at https://github.com/hulianyuyy/TVI-CoT.

## 1. Introduction

Chain-of-thought (CoT) reasoning has emerged as a transformative technique for enhancing problem-solving in large language models (Wei et al., 2022; Kojima et al., 2022). By decomposing complex problems into intermediate steps, CoT enables models to tackle tasks intractable through direct answer generation. The success of CoT has motivated its extension to multimodal LLMs (MLLMs) (Zhang et al., 2023; Xu et al., 2025; Shao et al., 2024). However, a fundamental limitation persists: *current multimodal CoT approaches perform reasoning entirely in text, without accessing visual features during the reasoning process*.

Consider how existing MLLMs handle a geometry problem. The image is encoded into a fixed representation projected into the language model's embedding space. From this point, all reasoning occurs in explicit text form in the CoT process. The model describes what it "sees," performs calculations, and derives conclusions, but cannot look back at the image to verify observations, extract additional details, or attend to different regions as reasoning evolves. This "**vision-blind reasoning**" forces MLLMs to rely entirely on whatever visual information was captured initially, regardless of whether it is sufficient for the task.

This contrasts sharply with human visual reasoning. When solving complex visual problems, humans engage in iterative looking, thinking, and re-examining. A mathematician solving geometry might identify given angles, reason about relationships, return to the diagram to locate auxiliary lines, and verify results against visual constraints. Current MLLMs lack ability for such iterative visual consultation.

The consequences are significant: (1) **fine-grained information is lost**. Subtle details relevant later are not captured initially; (2) **errors compound without correction**. misinterpretations cannot be verified against the image; (3) **different steps cannot attend to different regions**. Complex problems require extracting information from multiple regions at different stages. These limitations are pronounced in mathematical solving and complex reasoning, where problems require extracting multiple pieces of information from input images at different solution stages.

To handle this limitation, we propose **Text-Visual Interleaved Chain-of-Thought (TVI-CoT)**, a framework that addresses vision-blind reasoning through explicit interleaving of textual reasoning and visual feature access in the reasoning process. Our key insight is to introduce visual features interleaved with texts within the reasoning process, en-

---

[1]College of Computing and Data Science of Nanyang Technological University, Singapore. Correspondence to: Zeqin Liao <zeqin.liao@ntu.edu.sg>.

*Proceedings of the 43rd International Conference on Machine Learning*, Seoul, South Korea. PMLR 306, 2026. Copyright 2026 by the author(s).

abling the model to recap and recapture critical details from encoded input images to enhance the generation reliability. Specifically, TVI-CoT introduces learnable control tokens (⟨Think⟩, ⟨Look⟩, ⟨Answer⟩) to allow switching between textual reasoning and visual grounding. When producing ⟨Think⟩ token, the model starts explicit textual reasoning to generate chain-of-thoughts. Once the ⟨Look⟩ token is triggered, the model learns to focus on relevant image regions to extract fresh visual information for subsequent steps. This procedure is repeated until the ⟨Answer⟩ token is generated to offer the final answer.

Extensive results on commonly-used multimodal benchmarks verify that introducing visual evidence into the reasoning process instead of text-only thoughts notably improves the performance of MLLMs. We also notice that in complex scenarios such as math-solving problems, MLLMs typically require re-attending to the input images more than once to recap and model critical visual details to enhance reasoning confidence. We also provide comprehensive visualizations to verify that MLLMs are able to locate informative spatial regions to gain beneficial information.

## 2. Related Work

### 2.1. Multimodal Large Language Models

Early MLLMs such as Flamingo (Alayrac et al., 2022), BLIP-2 (Li et al., 2023), and LLaVA (Liu et al., 2023) established foundational architectures connecting visual encoders with language models through cross-attention, learned query tokens, and visual instruction tuning. The field has witnessed remarkable progress recently with increasingly capable models. InternVL series (Chen et al., 2024b; Zhu et al., 2025) introduce native multimodal pre-training and advance post-training techniques including test-time scaling and tool usage, achieving leading performance across diverse benchmarks. Qwen-VL series (Bai et al., 2025b;a) has evolved rapidly, with Qwen3-VL (Bai et al., 2025a) achieving state-of-the-art performance through architectural innovations. Kimi-VL (Team et al., 2025) proposes a mixture-of-experts architecture with efficient long-context processing. Ovis2 (Lu et al., 2025) advances visual embedding through hierarchical visual token generation, enabling more effective visual representation learning. Proprietary models have also advanced significantly. GPT-5 (Singh et al., 2025) demonstrates native multimodal understanding with significantly improved speed and capability. Claude-4.1-Opus (Anthropic, 2025) achieves strong performance on visual reasoning tasks with enhanced instruction following. Gemini (Comanici et al., 2025) introduces significantly improved multimodal capabilities with native tool use and agentic behaviors.

Despite this progress, most MLLMs process visual infor-

mation in a single forward pass and reason entirely in text, unable to re-access visual features during reasoning. Our work addresses this limitation through explicit mechanisms for repeated visual consultation.

### 2.2. Chain-of-Thought Reasoning

Chain-of-thought prompting, introduced by Wei et al. (Wei et al., 2022), has revolutionized the reasoning capabilities of LLMs. The key insight is to break complex problems into intermediate steps and allow models to allocate computation appropriately and avoid compounding errors. Zero-shot CoT (Kojima et al., 2022) demonstrated that simply prompting models to "think step by step" can elicit reasoning abilities without task-specific examples. Subsequent work explored self-consistency (Wang et al., 2022) through sampling multiple reasoning paths, tree-of-thought (Yao et al., 2023) for branching exploration, and program-of-thought (Chen et al., 2022) for executable code generation.

The reasoning paradigm was transformed when OpenAI's o1 (OpenAI, 2024) demonstrates that extended chain-of-thought reasoning with reinforcement learning can dramatically improve performance on complex mathematical and scientific problems. This sparked rapid development of open-source reasoning models. DeepSeek-R1 (Guo et al., 2025) achieves comparable reasoning capabilities through pure reinforcement learning without supervised fine-tuning on reasoning traces. Recently, several models have further advanced this paradigm. Skywork-OR1 (He et al., 2025) demonstrates effective open reasoning through carefully designed training pipelines. Marco-o1 (Zhao et al., 2024) explores open reasoning in multilingual settings. s1 (Muennighoff et al., 2025) introduces test-time scaling techniques that improve reasoning without additional training.

In the multimodal domain, Visual-CoT (Shao et al., 2024) introduces visual chain-of-thought that explicitly grounds reasoning steps in image regions through bounding box annotations. LLaVA-CoT (Xu et al., 2025) demonstrates that training MLLMs on structured reasoning data can significantly improve visual reasoning performance. Insight-V (Dong et al., 2025) explored long chain-of-thought generation for complex visual understanding. More recently, Visual-R1 (Huang et al., 2026) applies reinforcement fine-tuning to visual reasoning, achieving strong performance on mathematical and scientific tasks. Mulberry (Yao et al., 2024) proposes collective Monte Carlo tree search for multimodal reasoning, enabling exploration of multiple visual reasoning paths. VL-Rethinker (Wang et al., 2025) introduces selective sample replay and triggers the model to include self-reflection reasoning step during reasoning. VAPO-Thinker (Tian et al., 2025) lets the model to dynamically verify the correctness of related vision clues in input images to enhance reasoning ability.

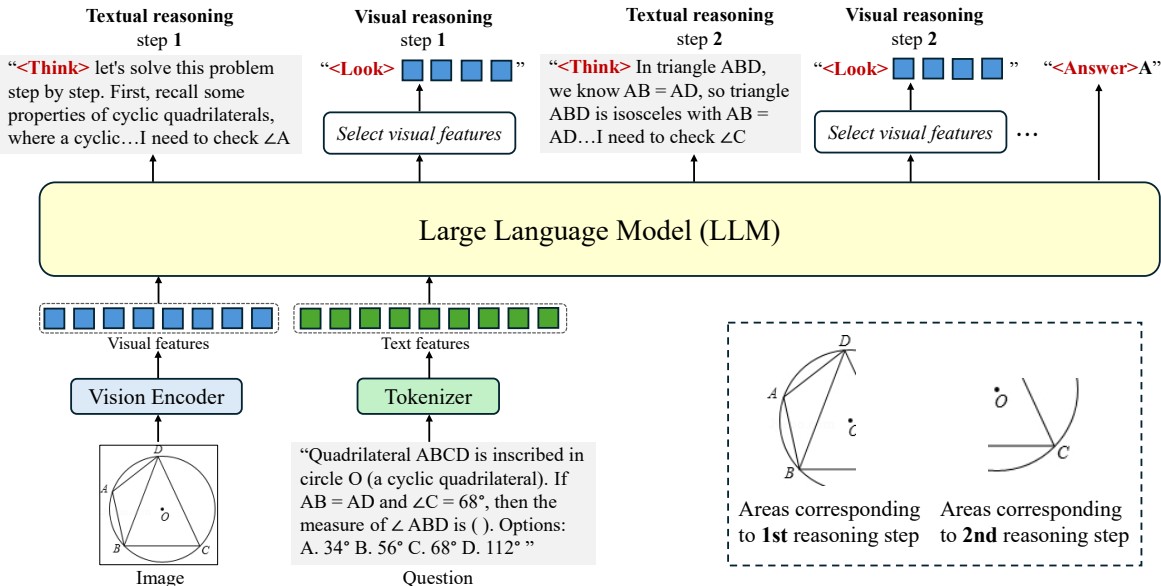

*Figure 1.* Overview of the TVI-CoT framework. Given an image and a question, the vision encoder and tokenizer produce visual and text features that are fed into the LLM. The model dynamically alternates between *textual reasoning steps* (⟨Think⟩), which perform chain-of-thought analysis, and *visual reasoning steps* (⟨Look⟩), which select relevant visual features from the image. Each visual reasoning step attends to different image areas conditioned on the current reasoning state (e.g., the 1st step focuses on $\angle A$ while the 2nd step shifts to $\angle C$). This interleaving continues adaptively until the model produces ⟨Answer⟩.

However, these approaches, even advanced models like GPT-5 or Gemini, perform CoT in pure text space. Once visual features are initially encoded, they cannot be re-accessed during reasoning. Our TVI-CoT framework addresses this by enabling dynamic, reasoning-conditioned visual attention that evolves with the problem-solving process.

### 2.3. Reasoning with Images

Recent advances in 'thinking with images' enhance multimodal reasoning by editing or revisiting images with external tools to integrate visual tokens into reasoning processes. Chen et al. (Chen et al., 2025) propose Mint-CoT for interleaved visual tokens in math reasoning, while Jiang et al. (Jiang et al., 2025) introduce VLM-R3 for improved reasoning via region recognition and refinement. Deep-Eyes (Zheng et al., 2026) enables the model to utilize image editing tools without supervised fine-tuning. UniVG-R1 (Bai et al., 2025c) allows the model to perform pixel-level grounding and greatly increase visual perception ability. V-Thinker (Qiao et al., 2025) presents a generative framework to incorporate generated images as internal representations for reasoning. OpenThinkIMG (Su et al., 2025) proposes a unified framework to plan sequential visual tool usage to support dynamic and explainable visual reasoning. However, 'thinking with images' requires external tools for image editing, which inevitably increase operating complexity and computational overhead. Instead, TVI-CoT could dynamically revisit input images within CoT, eliminating the need of time-intensive external operations.

## 3. Methodology

### 3.1. Problem Formulation

We consider multimodal reasoning tasks where the input consists of an image $I$ and a textual query $Q$, and the goal is to produce an answer $A$ along with a reasoning chain $R = \{r_1, r_2, \ldots, r_T\}$ that explains the solution process. Each reasoning step $r_t$ consists of a *textual reasoning* component $r_t^{\text{text}}$ and an optional *visual reasoning* component $r_t^{\text{vis}}$ that selects relevant visual features from the image. This formulation captures the iterative nature of human visual reasoning, where problem-solving involves repeated alternation between cognitive analysis and visual inspection.

Formally, let $\mathbf{v} = f_{\text{enc}}(I) \in \mathbb{R}^{N \times d}$ denote the visual features extracted by a vision encoder, where $N$ is the number of visual tokens and $d$ is the feature dimension. Let $\mathbf{q} = g_{\text{enc}}(Q) \in \mathbb{R}^{M \times d}$ denote the encoded query tokens. Our objective is to learn a model that generates:

$$P(A, R|I, Q) = \prod_{t=1}^{T} P(r_t|r_{<t}, \mathbf{v}, \mathbf{q}) \cdot P(A|R, \mathbf{v}, \mathbf{q}) \quad (1)$$

The key distinction from prior multimodal CoT approaches is that each reasoning step $r_t$ can dynamically modulate attention over the visual features $\mathbf{v}$, rather than treating them as a fixed input. This enables the model to extract

different visual information at different reasoning stages, mirroring the iterative visual consultation that characterizes human problem-solving.

## 3.2. Framework Overview

As illustrated in Figure 1, TVI-CoT builds on a standard MLLM architecture comprising a vision encoder, a text tokenizer, and a large language model, but augments it with a *dynamic text-visual interleaving* mechanism that enables the model to alternate between textual reasoning and visual reasoning at each step of the chain-of-thought process.

Concretely, the framework operates as follows. The image $I$ is encoded by the vision encoder into visual features $\mathbf{v}$, and the question $Q$ is converted into text features by the tokenizer. Both are fed into the LLM, which then generates a sequence of interleaved steps. In each *textual reasoning step*, the model emits a $\langle \text{Think} \rangle$ token followed by natural language analysis (e.g., recalling properties, forming hypotheses, or performing calculations). When the model determines that visual evidence is needed, it emits a $\langle \text{Look} \rangle$ token to trigger a *visual reasoning step*, in which a grounding module selects a subset of visual features relevant to the current reasoning context. Crucially, the selected visual features differ across steps, as shown in Figure 1, the first visual reasoning step attends to image areas corresponding to one aspect of the problem (e.g., $\angle A$), while the second step shifts attention to a different region (e.g., $\angle C$), enabling progressive extraction of the visual information required at each stage. This textual–visual alternation repeats adaptively until the model generates an $\langle \text{Answer} \rangle$ token to produce the final answer.

## 3.3. Visual Grounding Module

The visual grounding module dynamically selects relevant visual features based on the current reasoning state. At each visual reasoning step $t$, we compute attention weights over visual tokens using a dedicated grounding head:

$$\alpha_t = \text{softmax}\left( \frac{\mathbf{h}_t \mathbf{W}_q (\mathbf{v}\mathbf{W}_k)^\top}{\sqrt{d}} \right) \tag{2}$$

where $\mathbf{h}_t \in \mathbb{R}^d$ is the hidden state encoding the reasoning context up to step $t$, and $\mathbf{W}_q, \mathbf{W}_k \in \mathbb{R}^{d \times d}$ are learnable projection matrices. Because $\mathbf{h}_t$ aggregates all previous textual reasoning and grounding history, the attention is *reasoning-conditioned*: the model attends to different image areas at different steps depending on what information is currently needed.

The attended visual representation is computed as:

$$\mathbf{v}_t^{\text{att}} = \sum_{i=1}^{N} \alpha_{t,i} \cdot \mathbf{v}_i \mathbf{W}_v \tag{3}$$

where $\mathbf{W}_v \in \mathbb{R}^{d \times d}$ is a value projection matrix. This representation is injected back into the LLM context, providing the subsequent textual reasoning step with fresh visual evidence from the selected regions.

To enable interpretable grounding, we retain the top-$k$ visual tokens with highest attention weights as the grounding set $\mathcal{G}_t = \{i : \alpha_{t,i} \in \text{top-}k(\alpha_t)\}$, which can be mapped back to spatial image regions for visualization. We use $k = 32$ by default.

## 3.4. Dynamic Text-Visual Interleaving Mechanism

The core of TVI-CoT is the dynamic interleaving mechanism that coordinates the alternation between textual and visual reasoning through learnable control tokens. We introduce three special tokens, i.e., $\langle \text{Think} \rangle$, $\langle \text{Look} \rangle$, and $\langle \text{Answer} \rangle$, into the vocabulary to denote textual reasoning, visual reasoning, and answer generation modes respectively. The model's generation at step $t$ depends on which control token is predicted:

$$r_t = \begin{cases} r_t^{\text{text}} & \text{if mode} = \langle \text{Think} \rangle \\ r_t^{\text{vis}} = \mathcal{G}_t & \text{if mode} = \langle \text{Look} \rangle \\ A & \text{if mode} = \langle \text{Answer} \rangle \end{cases} \tag{4}$$

The control token prediction is performed by the LLM as part of its autoregressive generation, so the interleaving pattern is learned end-to-end. This design allows the model to *adaptively* determine when and how often to consult visual information. Simple problems may require only one $\langle \text{Think} \rangle$–$\langle \text{Look} \rangle$ pair before $\langle \text{Answer} \rangle$, while complex problems trigger multiple rounds of textual–visual alternation (as depicted in Figure 1). We initialize the control token embeddings from semantically related words ("think," "look," "answer") and apply a mild regularization to prevent degenerate patterns where the model either never grounds or grounds excessively.

## 3.5. Reasoning Chain Composition

Given the selected visual features, we inject them directly into the output reasoning sequence. At each $\langle \text{Look} \rangle$ step, the attended representation $\mathbf{v}_t^{\text{att}}$ from the grounding module (Section 3.3) is linearly projected into the LLM's embedding space and concatenated with the current hidden-state context:

$$\mathbf{h}_{t+1} = \text{LLM}\left( [\mathbf{h}_{<t}; \ \mathbf{W}_{\text{proj}} \mathbf{v}_t^{\text{att}}] \right) \tag{5}$$

where $\mathbf{W}_{\text{proj}} \in \mathbb{R}^{d \times d}$ is a learnable projection matrix. The projected features act as implicit visual tokens that provide subsequent $\langle \text{Think} \rangle$ steps with direct access to freshly grounded visual evidence.

The resulting reasoning chain therefore alternates between textual tokens produced during $\langle \text{Think} \rangle$ and visual feature

tokens injected during ⟨Look⟩. The ⟨Think⟩ token triggers textual reasoning which mimics the human thinking process with explicit text outputs. The ⟨Look⟩ token triggers visual reasoning process which provides visual evidence and potential support for the textual thoughts. This loop is repeated until a ⟨Answer⟩ token is omitted to give the final answer. Because all grounded regions are recorded, the full trace remains interpretable and verifiable.

### 3.6. Training Objective

We train TVI-CoT using a combination of supervised learning on annotated reasoning chains and an auxiliary grounding loss that encourages accurate visual attention. The supervised loss follows standard autoregressive language modeling:

$$\mathcal{L}_{\text{SFT}} = -\sum_{t=1}^{T} \log P(r_t | r_{<t}, \mathbf{v}, \mathbf{q}) \tag{6}$$

This loss trains the model to generate appropriate control tokens, reasoning text, and final answers based on the annotated examples.

To encourage effective visual grounding that attends to relevant image regions, we introduce an auxiliary grounding loss:

$$\mathcal{L}_{\text{ground}} = -\sum_{t \in \mathcal{T}_{\text{look}}} \sum_{i \in \mathcal{G}_t^*} \log \alpha_{t,i} \tag{7}$$

where $\mathcal{T}_{\text{look}}$ denotes steps with grounding operations and $\mathcal{G}_t^*$ is the ground-truth grounding set derived from annotations (bounding boxes or region labels associated with each reasoning step).

For samples without explicit grounding annotations, we use a weak supervision signal based on object detection results or attention from a pretrained grounding model, providing approximate guidance for the visual attention mechanism.

The final training objective combines both losses:

$$\mathcal{L} = \mathcal{L}_{\text{SFT}} + \lambda \mathcal{L}_{\text{ground}} \tag{8}$$

where $\lambda$ is a hyperparameter balancing the two objectives. In our experiments, we find $\lambda = 0.1$ provides a good balance between reasoning quality and grounding accuracy.

### 3.7. Training Data Construction

Obtaining training data with annotated interleaved reasoning chains is challenging, as existing datasets typically provide only final answers or purely textual explanations. We construct training data through two approaches:

**Synthetic generation.** We synthesize the dataset using LLaVA-178k by asking Gemini3-Pro to determine if visual evidence is needed and generate text-visual CoT chains when applicable. These annotations are validated by Qwen3-VL, resulting in 55k samples. Finally, we conduct a quality check where Gemini3-Pro and Qwen3-VL-235B-A22B achieve over 97% agreement, indicating high data annotation quality. Two human experts are recruited to verify 5% samples manually, and an overall acceptance rate of 98.7% is reached. We therefore use this semi-annotated dataset for training.

**Grounding augmentation.** We augment our training data from two major visual CoT datasets: Visual-CoT and Zebra-CoT. For Visual-CoT, which includes auxiliary bounding boxes, we use Gemini3-Pro to verify whether visual annotations are necessary, whether existing boxes are relevant, and whether additional annotations are needed. When required, Gemini3-Pro generates new bounding boxes, which are further validated by Qwen3-VL-235B-A22B for logical correctness and spatial precision. After this semi-automatic process, 52k samples are retained with updated CoT chains. For Zebra-CoT, we keep samples from 2D visual reasoning tasks, and for the rest, Gemini3-Pro checks annotation correctness and if additional annotations are needed, yielding 43k refined samples.

This combined approach yields about 150K training examples with interleaved reasoning and grounding annotations.

## 4. Experiments

### 4.1. Experimental Setup

**Implementation details.** We initialize TVI-CoT with Qwen3-VL-8B-Instruct (Bai et al., 2025a) and continue training on our interleaved reasoning dataset. Training proceeds for 3 epochs with a learning rate of 2e-5 (with cosine decay), batch size of 8, and AdamW optimizer with weight decay 0.001. The grounding loss weight $\lambda$ is set to 0.1 based on validation performance. We use $k = 32$ for top-$k$ grounding selection, providing a balance between spatial precision and computational efficiency. All experiments are conducted on 8 NVIDIA A100 80GB GPUs, with total training time of 36 hours.

**Benchmarks and baselines.** We evaluate TVI-CoT on 8 challenging multimodal benchmarks that collectively cover a broad range of visual reasoning capabilities including MMMU (Yue et al., 2024), MMBench (Liu et al., 2024),MathVerse (Zhang et al., 2024), MathVista (Lu et al., 2023), ScienceQA (Lu et al., 2022), MMStar (Chen et al., 2024a), AI2D (Kembhavi et al., 2016) and MMT-Bench (Ying et al., 2024). We compare against state-of-the-art MLLMs and MLLM-based CoT methods. For general MLLMs, we include proprietary models GPT-5-high (Singh et al., 2025), Claude-Opus-4.1 (Anthropic, 2025) and Gemini-2.5-Pro (Comanici et al., 2025), and open-source MLLMs including LLaVA-Onevision-1.5 (An et al.,

*Table 1.* Main results on multimodal benchmarks. Best results among methods with similar model scale in **bold**. For Qwen3-VL-Instruct-8B, we report our reproduced results under the same setting.

| Model | MMMU | MMBench | MathVerse | MathVista | ScienceQA | MMStar | AI2D | MMT-Bench |
|---|---|---|---|---|---|---|---|---|
| *Proprietary Models* | | | | | | | | |
| GPT-5-high | 84.4 | 83.8 | 84.1 | 81.3 | 97.4 | 76.4 | 89.7 | 77.2 |
| Claude-Opus-4.1 | 77.2 | 83.0 | 68.1 | 74.5 | - | 71.0 | 84.4 | - |
| Gemini-2.5-Pro | 81.7 | 90.1 | - | 82.7 | 96.2 | 77.5 | 88.4 | 75.4 |
| *Open-Source MLLMs (7-8B scale)* | | | | | | | | |
| LLaVA-OneVision-1.5-8B | 55.4 | 84.1 | - | 69.6 | 95.0 | 67.7 | 84.2 | - |
| InternVL3-8B | 62.7 | 83.4 | 39.8 | 71.6 | - | 68.2 | 85.2 | 65.0 |
| *MLLM-based Chain-of-Thought Methods* | | | | | | | | |
| LLaVA-CoT | - | 75.0 | 43.2 | 54.8 | - | 57.6 | 78.7 | - |
| Insight-V | 50.2 | 82.3 | - | 59.9 | 61.5 | 79.8 | - | - |
| Mulberry | 55.0 | - | - | 63.1 | - | 61.3 | - | - |
| Vision-R1-7B | - | - | 52.4 | 73.5 | - | - | - | - |
| VL-Rethinker-7B | 56.7 | - | 54.2 | 74.9 | - | 62.7 | - | - |
| VAPO-Thinker-7B | 60.2 | - | 53.3 | 75.6 | 92.0 | 68.5 | 82.3 | 64.1 |
| Qwen3-VL-8B (Baseline) | 58.2 | 83.4 | 56.4 | 73.2 | 91.8 | 67.1 | 83.6 | 63.3 |
| **TVI-CoT (Ours)** | **64.3** (+6.1%) | **84.7** (+1.3%) | **60.2** (3.8%) | **76.6** (+3.4%) | **95.2** (+3.4%) | **69.8** (+2.7%) | **85.6** (+2.0%) | **65.4** (+2.1%) |

2025) and InternVL3 (Zhu et al., 2025). For MLLM-based CoT methods, we compare against (1) Visual-CoT (Shao et al., 2024) which explicitly asks the model to provide visual regions with bounding boxes, and (2) LLaVA-CoT (Xu et al., 2025) which trains on structured reasoning data with summary, caption, reasoning and conclusion steps, and (3) Insight-V (Dong et al., 2025) which explores long chain-of-thought generation, and (4) Mulberry (Yao et al., 2024) which uses Monte Carlo tree search for multimodal reasoning, and (5) Vision-R1 (Huang et al., 2026) which introduces GRPO-based reinforcement learning to boost the reasoning ability of MLLMs, and (6) VL-Rethinker (Wang et al., 2025) which presents selective sample replay and rethinking trigger strategies to enforce a self-reflection reasoning step, and (7) VAPO-Thinker (Tian et al., 2025) which prompts the model to verify if the relied vision clues are correct in the images. However, in the generated thoughts, existing methods either lacks the ability to perform explicit visual reasoning or restrict the model to perform visual grounding only once regardless of circumstances. None of them enables the model to dynamically switch between textual reasoning and visual reasoning with adaptive steps.

### 4.2. Main Results

Table 1 presents the main results across all benchmarks. Compared to open-source models and MLLM-based CoT methods with comparable scales, TVI-CoT achieves state-of-the-art performance. On MMMU, TVI-CoT achieves 64.3%, outperforming the previous best MLLM-based CoT method VAPO-Thinker (Tian et al., 2025) by 4.6% and the backbone Qwen3-VL-8B (Bai et al., 2025a) by 6.1%. On mathematical reasoning benchmarks, TVI-CoT shows particularly substantial improvements. On MathVerse, TVI-CoT achieves 60.2%, a 6.9% improvement over the previous

*Table 2.* Detailed results on mathematical benchmarks by problem types. Improvements over Qwen3-VL-8B shown in parentheses.

| Category | LLaVA-CoT | Qwen3-VL-8B | TVI-CoT |
|---|---|---|---|
| *MathVerse* | | | |
| All | 43.2 | 56.4 | **60.2 (+3.4)** |
| Plane Geometry | 46.2 | 60.4 | **66.7 (+6.3)** |
| Solid Geometry | 39.1 | 51.8 | **58.3 (+6.5)** |
| Functions | 42.0 | 54.2 | **57.2 (+2.8)** |
| *MathVista* | | | |
| All | 54.8 | 73.2 | **76.6 (+3.4)** |
| Figure question answering | 48.6 | 66.2 | **73.2 (+7.0)** |
| Geometry problem solving | 55.4 | 74.8 | **80.5 (+5.7)** |
| Math word problem | 58.6 | 52.6 | **54.8 (+2.2)** |
| Text-book question answering | 69.2 | 84.1 | **85.4 (+1.3)** |
| Visual question Answering | 41.4 | 58.3 | **62.5 (+4.2)** |

best VAPO-Thinker (Tian et al., 2025) and 3.8% improvement over the backbone. On MathVista, TVI-CoT reaches 76.6%, outperforming all MLLM-based methods and even beating Claude-Opus-4.1 (74.5%). These results demonstrate that iterative visual grounding is especially effective for problems requiring multi-step mathematical reasoning with visual diagrams, where repeated consultation of visual features is critical. On other benchmarks, TVI-CoT demonstrates consistent superiority, which outperforms all competitive MLLMs and previous CoT-based methods and achieves >2% performance boost on most benchmarks compared to the backbones. These results confirm that the interleaved reasoning mechanism generalizes effectively across scientific, visual, and comprehensive multimodal reasoning tasks, consistently outperforming methods that perform CoT reasoning purely in the textual space.

### 4.3. Analysis on Mathematical Reasoning

We conduct detailed analysis on MathVerse and MathVista to understand the source of improvements. Table 2 breaks

*Table 3.* Category breakdown results on MathVerse. Improvements over Qwen3-VL-8B shown in parentheses.

| Category | Qwen3-VL-8B | TVI-CoT |
|---|---|---|
| Text Dominant | 66.5 | **68.7** (**+2.2**) |
| Text Lite | 58.7 | **60.4** (**+1.7**) |
| Text Only | 61.2 | **65.0** (**+3.8**) |
| Vision Intensive | 52.9 | **58.0** (**+5.1**) |
| Vision Dominant | 50.4 | **54.6** (**+4.2**) |
| Vision Only | 46.2 | **52.4** (**+6.2**) |

*Table 4.* Performance of TVI-CoT on Qwen3-VL-32B.

| Methods | MMMU | MMBench | MathVista | MathVerse | MMStar |
|---|---|---|---|---|---|
| Qwen3-VL-32B | 76.0 | 87.6 | 83.8 | 76.8 | 77.7 |
| TVI-CoT | **78.2** (**+2.2**) | **88.6** (**+1.0**) | **85.6** (**+1.8**) | **78.8** (**+2.0**) | **79.2** (**+1.5**) |

*Table 5.* Ablation study on MMMU and MathVista benchmarks. Each row removes or modifies one component from the full TVI-CoT model.

| Configuration | MMMU | MathVerse |
|---|---|---|
| Qwen3-VL-8B (backbone) | 58.2 | 56.4 |
| TVI-CoT | **64.3** | **60.2** |
| w/o Interleaving (text→ visual) | 56.6 | 55.2 |
| w/o Interleaving (visual→ text) | 58.9 | 57.2 |
| w/o Grounding Loss | 62.4 | 58.6 |
| Random grounding regions | 51.2 | 52.3 |

down performance by problem category, revealing where interleaved reasoning provides the greatest benefit.

We notice that TVI-CoT achieves the largest improvements on problems with highly visual demand, where iterative visual grounding is most beneficial. On plane geometry and solid geometry problems of MathVista, we observe a +6.3% & 6.5% improvement over the Qwen3-VL-8B baseline. Experimental results on the figure question answering and geometry problem solving subsets also show 7.0% & 5.7% performance improvement. These results strongly support our hypothesis that problems requiring repeated consultation of visual elements benefit most from the interleaved reasoning mechanism. In contrast, functions and statistics problems show smaller but still substantial improvements (2.8% on MathVerse for functions, 2.2% on MathVista for math problem solving). We attribute this to these problem types often requiring extraction of data from charts or graphs, which can sometimes be accomplished with fewer visual grounding steps compared to complex geometric reasoning.

### 4.4. Category Breakdown Results on MathVerse

We evaluate TVI-CoT and Qwen3-VL-8B with lmms-eval and list the category breakdown results in Tab. 3. From text dominant category to vision only category, Qwen3-VL exhibits an overall descending trend on performance, demonstrating its weak ability to excavate beneficial information from visual-dominant inputs. In contrast, compared to Qwen3-VL, TVI-CoT brings considerable performance boost across all categories, demonstrating its effectiveness on visual reasoning. Especially, TVI-CoT achieves more performance boost on visual-biased categories (+5.2% on average for Vision Intensive, Vision Dominant and Vision Only) that text-biased categories (+2.6% on average for Text Dominant, Text Lite and Text Only). This shows that TVI-CoT can more effectively capture key information from visual inputs, wherein the adaptive revisiting mechanism plays a crucial role to incorporate visual cues within CoT to provide better visual evidence.

### 4.5. Performance on Larger Models

We implement TVI-CoT based on Qwen3-VL-32B and report the performance in Tab. 4. Due to memory constraint,

we are only able to train Qwen3-VL-32B with LoRA on 8*80G GPUs. Compared to Qwen3-VL-32B, TVI-CoT could notably improve the performance on various benchmarks with +1.7% on average. Notably, it brings + 1.8% and +2.0% performance boost on math benchmarks (MathVista, MathVerse). This shows that TVI-CoT could well generalize to larger models, and demonstrates that the dynamic token-based revisiting mechanism is the key factor to improve visual reasoning ability.

### 4.6. Ablation Studies

We conduct comprehensive ablation studies to analyze the contribution of each component. Table 5 shows results on the MMMU and MathVerse benchmarks.

**Interleaving is critical.** Removing the interleaving mechanism (generating all reasoning either before or after visual grounding) leads to a substantial drop on MMMU and MathVerse, confirming that iterative visual-textual interaction is essential for complex reasoning.

**Grounding supervision helps.** The grounding loss contributes 1.9% on MMMU and 3.0% on MathVerse. Without explicit grounding supervision, the model learns to generate grounding tokens but attends to less relevant image regions.

**Grounding quality matters.** Replacing learned grounding with random region selection leads to substantial degradation (-13.1% on MMMU, -7.9% on MathVista), confirming that accurate visual attention is essential for the interleaved mechanism to be effective.

### 4.7. Effect of Text-Visual Interleaving Steps

To understand how the number of visual grounding operations affects reasoning performance, we fix the interleaving steps from 0 to 4 and compare with our adaptive design. When the step equals 0, the model degenerates into pure

*Table 6.* Performance with fixed number of interleaving steps (0–4). Step 0 represents text-only reasoning without visual grounding. Adaptive denotes TVI-CoT that dynamically determines the number of steps per sample.

| Steps | 0 (Text-only) | 1 | 2 | 3 | 4 | Adaptive |
|---|---|---|---|---|---|---|
| **MMMU** | 58.2 | **62.3** | 61.2 | 59.6 | 56.4 | **64.3** |
| **MathVerse** | 56.4 | 58.2 | **58.8** | 57.2 | 55.8 | **60.2** |

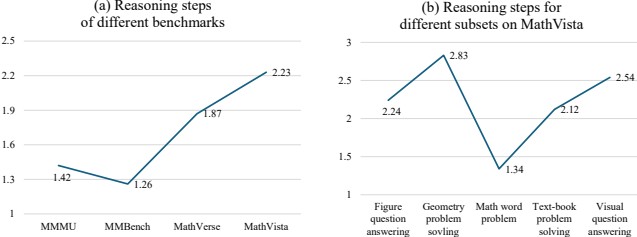

*Figure 2.* Analysis of grounding patterns across problem types and benchmarks. Problems with higher visual complexity (e.g., geometry) trigger more grounding operations, reflecting the multi-step visual analysis required for spatially demanding reasoning.

textual reasoning without any visual grounding during inference. Table 6 presents results on MMMU and MathVerse.

First, introducing even a single interleaving step (Step 1) yields clear gains over text-only reasoning: +4.1% on MMMU and +1.8% on MathVerse, confirming that accessing visual features during reasoning provides immediate benefits. Second, the optimal fixed step count differs across benchmarks, MMMU peaks at Step 1 (62.3%) while MathVerse peaks at Step 2 (58.8%), reflecting that different tasks demand varying degrees of visual consultation. Third, performance degrades when too many fixed steps are imposed (Step 3–4), as excessive visual grounding introduces redundant or noisy information that disrupts the reasoning flow. Most importantly, the adaptive mechanism (64.3%/60.2%) substantially outperforms all fixed-step configurations, demonstrating the necessity of dynamically deciding to perform visual grounding on a per-sample basis.

### 4.8. Analysis of Grounding Patterns

To better understand how TVI-CoT allocates visual attention, we analyze the grounding patterns produced by the adaptive interleaving mechanism. Figure 2 offers a full visualization for the distribution of ⟨Look⟩ operations across different problem types and benchmarks.

**Benchmark-level analysis.** Figure 2(a) shows that the model allocates more reasoning steps to visually complex benchmarks: MathVista averages the most (2.23), followed by MathVerse (1.87), MMMU (1.42), and MMBench (1.26). This ranking aligns with Table 6, where MathVerse benefits substantially from additional interleaving steps while MMMU plateaus earlier.

*Table 7.* Ablations for the number $k$ of retrieved visual tokens.

| $k$ | 0 | 8 | 16 | 32 | 64 |
|---|---|---|---|---|---|
| MMMU | 58.9 | 62.1 | 63.2 | **64.3** | 64.1 |
| MathVerse | 55.2 | 58.4 | 59.8 | **60.2** | 59.2 |

**Problem-type analysis.** Figure 2(b) breaks down the step distribution on MathVista by problem type. Geometry problem solving triggers the most ⟨Look⟩ operations (2.83), followed by visual question answering (2.54), figure question answering (2.24), and textbook problem solving (2.12), while math word problems require the fewest (1.34). This pattern is consistent with Table 2, where geometry subcategories exhibit the largest accuracy gains, confirming that TVI-CoT learns to invest more visual attention where it yields the greatest benefit.

### 4.9. Analysis for the number of retrieved visual tokens

We provide ablations for the number $k$ of retrieved visual tokens in Tab. 7. It's worth noting that when $k$ equals 0, the model would degenerate into pure textual reasoning. We observe that as $k$ increases from 0, the performance continues increasing at first and reaches a peach when equaling 32. Further increasing $k$ slightly hurts the performance, which may be due to the overly introduced visual redundancy. We thus set $k$ as 32 by default.

### 4.10. Qualitative Analysis

Figure 3 presents example reasoning chains generated by TVI-CoT, illustrating how the model interleaves textual reasoning with visual grounding during problem-solving.

In Figure 3(a), the model reasons about the consequence of algae dying in a food web. Step 1 identifies the need to locate algae and its role as a primary producer via ⟨Think⟩, then ⟨Look⟩ attends to the algae, plankton, and seaweed region. Step 2 reasons that dependent organisms will lose their food source, and ⟨Look⟩ shifts to the consumers including mussel, crab, and limpet to verify the trophic chain before producing ⟨Answer⟩. In Figure 3(b), the model solves a $3 \times 3$ matrix puzzle. In Step 1, it first identifies the overall matrix layout and recognizes the task as a pattern-completion problem. The ⟨Look⟩ operation is then directed to the top row to examine the visual relationships among elements to uncover a potential composition rule. Step 2 hypothesizes that the underlying rule is (column 1 + column 3 = column 2) and ⟨Look⟩ shifts to the middle row for verification, then confidently selects the answer.

Both examples demonstrate that the attended regions shift meaningfully between grounding steps, driven by specific information needs articulated in each ⟨Think⟩ phase, producing transparent and verifiable reasoning traces.

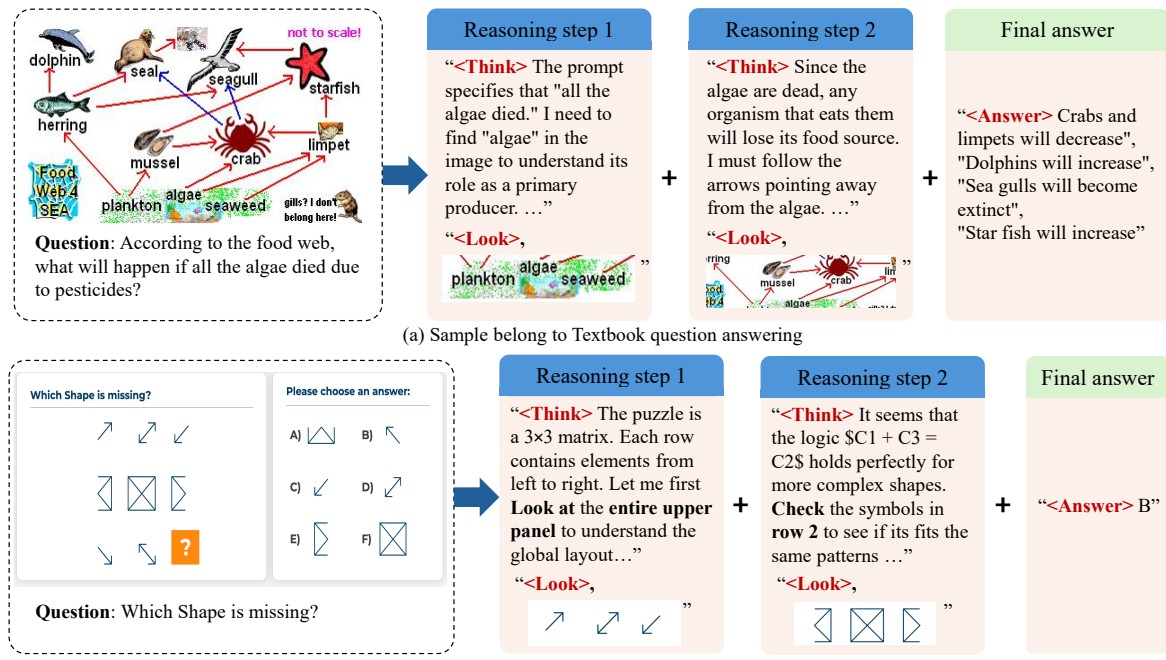

*Figure 3.* Qualitative examples of TVI-CoT's interleaved reasoning. The model alternates between textual reasoning (⟨Think⟩) and visual grounding (⟨Look⟩), producing interpretable chains with explicit visual references at each step.

*Table 8.* Efficiency comparison on MathVerse. All metrics are relative to the Qwen3-VL-8B backbone (1.0×).

| Method | Accuracy | Latency | Memory | Output length |
|---|---|---|---|---|
| Qwen3-VL-8B | 56.4% | 1.0× | 1.0× | 1.0× |
| Visual-CoT | 43.2% | 1.16× | 1.25× | 0.85× |
| TVI-CoT | 60.2% | 1.12× | 1.10× | 1.05× |

### 4.11. Efficiency Analysis

We analyze the computational overhead introduced by the interleaving mechanism. Table 8 compares inference efficiency across methods.

As shown in Table 8, TVI-CoT achieves notable accuracy gain over the backbone with only 12% additional latency and 10% memory overhead, since visual features are cached and reused without recomputation. Compared to Visual-CoT, TVI-CoT is both more accurate (+17.0%) and more efficient in latency (1.12× vs. 1.16×) and memory (1.10× vs. 1.25×), as Visual-CoT requires explicit bounding box generation and region cropping. The output length is only marginally longer than the backbone (1.05×), and the latency scales adaptively with reasoning complexity, correlating computation with problem difficulty.

### 4.12. Grounding Quality Analysis

To assess the quality of the visual regions selected by TVI-CoT's grounding module, we conduct an oracle experiment.

At each ⟨Look⟩ step, we replace the model's predicted regions with ground-truth regions annotated by human experts, keeping the rest of the pipeline unchanged. On 100 selected samples, we notice that replacing the predicted grounding regions with oracle regions yields modest improvements of 3.0%. The slight gap is expected, as geometry and diagram problems often contain spatially dense elements (e.g., overlapping labels, auxiliary lines) where precise region selection is more critical. Notably, TVI-CoT has substantially achieved considerable performance boost by dynamically grounding visual elements (e.g., +6.1% on MMMU), which confirms that the grounding module already learns a high-quality attention policy.

## 5. Conclusion

We presented TVI-CoT, a framework that addresses vision-blind reasoning in MLLMs by explicitly interleaving textual chain-of-thought steps with visual grounding operations through learnable control tokens (⟨Think⟩, ⟨Look⟩, ⟨Answer⟩). TVI-CoT achieves state-of-the-art results across eight benchmarks, with particularly strong gains on geometry-solving and figure-based problems where iterative visual consultation is most critical. Ablation studies confirm that the interleaving mechanism, grounding supervision, and adaptive step allocation each contribute meaningfully. We hope this work encourages future multimodal systems to move beyond single-pass visual encoding toward dynamic, reasoning-integrated visual processing.

## Acknowledgements

This research is supported by the National Research Foundation, Singapore, and DSO National Laboratories under the AI Singapore Programme (AISG Award No: AISG4-GC-2023-008-1B); by the National Research Foundation Singapore and the Cyber Security Agency under the National Cybersecurity R&D Programme (NCRP25-P04-TAICeN); and this research is part of the IN-CYPHER programme and is supported by the National Research Foundation, Prime Minister's Office, Singapore under its Campus for Research Excellence and Technological Enterprise (CREATE) programme. Any opinions, findings and conclusions, or recommendations expressed in these materials are those of the author(s) and do not reflect the views of the National Research Foundation, Singapore, Cyber Security Agency of Singapore, Singapore.

## Impact Statement

This paper presents work whose goal is to advance multimodal reasoning in machine learning. By improving interpretability through explicit ⟨Think⟩–⟨Look⟩ reasoning traces, TVI-CoT can benefit education, accessibility, and scientific discovery. As with any advance in multimodal AI, improved visual reasoning could be misused in surveillance or automated decision-making; deployment in sensitive domains (e.g., medical imaging) requires rigorous validation and human oversight. Our training data, derived from public VQA datasets and Gemini-generated chains, may encode source biases that future work should audit. The computational footprint is moderate (8B backbone, 12% inference latency overhead), and the adaptive mechanism avoids unnecessary computation on simpler problems.

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
