# OpenReview forum: "TVI-CoT: Text-Visual Interleaved Chain-of-Thought Reasoning for Multimodal Understanding"
_ICML.cc/2026/Conference — ICML 2026 regular_

### Official Review · Reviewer_baWM · 2026-03-06

**Soundness:** 3
**Presentation:** 3
**Significance:** 3
**Originality:** 2
**Overall Recommendation:** 4
**Confidence:** 4

**Summary:**

This paper aims to address the issue of "vision-blind reasoning" in existing MLLMs during chain-of-thought reasoning. The paper proposes the Text-Visual Interleaved Chain-of-Thought framework. This framework introduces three learnable control tokens into the vocabulary, allowing the model to adaptively switch between textual reasoning and dynamic visual grounding. Experiments demonstrate that TVI-CoT, initialized on Qwen3-VL-8B, achieves state-of-the-art results among MLLM-based CoT methods across 8 multimodal benchmarks.

**Compliance With Llm Reviewing Policy:**

Affirmed.

**Final Justification:**

My problem has been completely solved.

**Key Questions For Authors:**

Same as weaknesses.

**Limitations:**

No, need to discuss.

**Strengths And Weaknesses:**

### Strengths
- The proposed framework excellently mimics the iterative cognitive process of human visual reasoning, which involves "looking, thinking, and re-examining" when solving complex visual problems.
- Compared to baseline methods that rely on generating explicit bounding boxes (e.g., Visual-CoT) or re-cropping images, TVI-CoT proposes an extremely efficient implicit vector injection mechanism. It extracts information by computing attention weights over a reused pool of visual features.
- The paper conducts comprehensive ablation studies, strongly demonstrating the necessity of each core component.

### Weaknesses

- The authors claim that "current multimodal CoT approaches perform reasoning entirely in text, without accessing visual features during the reasoning process". However, a large body of similar work on "Thinking with Images" has recently emerged in the academic community (e.g., MINT-CoT[1], VLM-R³[2], etc.). The authors should provide a detailed discussion to supplement these missing related works.

- Although the reasoning steps of the model are adaptive, in the Visual Grounding Module, the authors force the retention of the top-32 (k=32) visual tokens with the highest attention weights as visual evidence. This hard-coded hyperparameter lacks dynamic adaptability. In practical scenarios, simple visual cues might only require a few tokens, whereas complex global chart structures might need significantly more than 32 tokens to be fully expressed.

- With the rapid development of the open-source community, fully open-source models like Bee[3] have recently emerged. I am very curious about the performance of the TVI-CoT framework on such models and the transferability of its training paradigm.

- The qualitative analysis section only presents successful cases where the model perfectly executes the text-visual interleaving. However, for such a multi-step interleaved system, error propagation is a critical issue. The paper completely lacks discussion on the following: if the model focuses on the wrong image region during a "Look" step and extracts incorrect features, does the subsequent "Think" step possess the ability for self-correction, or will this cause the model to generate hallucinations/nonsense?

[1] Chen, X., Zhang, R., Jiang, D., Zhou, A., Yan, S., Lin, W., & Li, H. (2025). Mint-cot: Enabling interleaved visual tokens in mathematical chain-of-thought reasoning. arXiv preprint arXiv:2506.05331.

[2] Jiang, C., Heng, Y., Ye, W., Yang, H., Xu, H., Yan, M., ... & Zhang, S. (2025). VLM-R $^ 3$: Region Recognition, Reasoning, and Refinement for Enhanced Multimodal Chain-of-Thought. arXiv preprint arXiv:2505.16192.

[3] Zhang, Y., Ni, B., Chen, X. S., Zhang, H. R., Rao, Y., Peng, H., ... & Hu, S. M. (2025). Bee: A High-Quality Corpus and Full-Stack Suite to Unlock Advanced Fully Open MLLMs. arXiv preprint arXiv:2510.13795.

---

> ### Author Rebuttal · Authors · 2026-03-31
>
> 1. Further discussion about works in "Thinking with Images".
>
> Thanks for your insightful suggestion! We acknowledge that “Thinking with Images” improves visual reasoning by leveraging external tools (e.g., zooming or image editing) to modify inputs and extract useful information. However, this approach requires sophisticated system design and additional tools (e.g., Python or other models), leading to increased computational and operational overhead. In contrast, TVI-CoT does not rely on external tools and can revisit input images via attention-based token selection, enabling an end-to-end design with lower complexity and improved efficiency. For discussion with “Thinking with Images,” we plan to include the following in Related Works:
>
> “Recent advances in ‘thinking with images’ enhance multimodal reasoning by editing or revisiting images with external tools to integrate visual tokens into reasoning processes. Chen et al. (2025) propose Mint-CoT for interleaved visual tokens in math reasoning, while Jiang et al. (2025) introduce VLM-R3 for improved reasoning via region recognition and refinement. DeepEyes (Zheng et al., 2025) enables the model to utilize image editing tools without supervised fine-tuning. UniVG-R1 (Bai et al., 2025) allows the model to perform pixel-level grounding and greatly increase visual perception ability. V-Thinker (Qiao et al., 2025) presents a generative framework to incorporate generated images as internal representations for reasoning. OpenThinkIMG (Su et al., 2025) proposes a unified framework to plan sequential visual tool usage to support dynamic and explainable visual reasoning. However, “thinking with images” requires external tools for image editing, which inevitably increase operating complexity and computational overhead. Instead, TVI-CoT could dynamically revisit input images within CoT, eliminating the need of time-intensive external operations.”
>
> 2. Adaptive design for the Visual Grounding Module
>
> Thanks for your interesting observation! Actually, we have tried adaptively deciding the top-k visual tokens with highest attention scores. However, we find this mechanism faces a practical problem that the model easily locates incoherent visual tokens distributed over various image regions. This hinders semantic interpretability for the revisiting operation, and practically leads to superior performance compared to hard-coded top-k selection (k=32). We paster the results here for your convenient reference.
>
> | Methods | MMMU | MMBench | MathVista | MathVerse |
> | --- | --- | --- | --- | --- |
> | Adaptive top-k | 62.2 | **84.8** | 59.1 | 74.8 |
> | Hard-coded top-k (k=32) | **64.3** | 84.7 | **60.2** | **76.6** |
>
> 3. Experiments on more open-source models.
>
> Thanks for your attention. We are also interested in whether TVI-CoT could transfer to more backbones with scalable effectiveness. However, due to the short time period during rebuttal, we struggle with completing the transferring process on the Bee model and haven’t finished that. We here provide the results of TVI-CoT based on Qwen2.5-VL-7B and Qwen3-VL-32B to verify its scalability across multiple backbones as follows. We observe that TVI-CoT successfully improves the visual reasoning ability of different backbones (+3.9% for Qwen2.5-VL-7B and +1.8% for Qwen3-VL-32B on average). It also works well upon a larger model (32B) with notable performance gain across different benchmarks. This demonstrates that TVI-CoT could notably revoke the visual perception ability via the dynamic revisiting mechanism. We are working on the Bee model and will provide the results once we have finished it ;).
>
> | Methods | MMMU | MMBench | MathVista | MathVerse |
> | --- | --- | --- | --- | --- |
> | Qwen2.5-VL 7B | 57.8 | 83.2 | 68.4 | 49.6 |
> | TVI-CoT | 63.2 (**+5.4%**) | 84.4 (**+1.2%**) | 73.4 (**+3.8%**) | 54.6 (**+5.0%**) |
> | Qwen3-VL-32B | 76.0 | 87.6 | 83.8 | 76.8 |
> | TVI-CoT | 78.2 (**+2.2%**) | 88.6 (**+1.0%**) | 85.6 (**+1.8%**) | 78.8 (**+2.0%**) |
>
> 4. Analysis for the potential error propagation in CoT.
>
> Thanks for your insightful question! We have several interesting observations for this question. We manually checked the reasoning chain of TVI-CoT, and found it does make errors during the LOOK execution by locating wrong image regions. Interestingly, it occasionally finds that it makes mistakes by outputting something like “Oh, I have located wrong areas before…” in the next THINK process. In this situation, it will correct the preceding mistake and make following plans. But sometimes, it doesn’t realize that it has made errors, and continue thinking with hallucinations and walk towards a wrong direction. This demonstrates that TVI-CoT possesses the ability to correct the mistakes during revisiting input images, but inherits the hallucination deeply rooted in LLMs. We believe parallel CoT or enforced reflection may better relieve the problem, and we will further work on this.

---

> > ### Author Rebuttal · Reviewer_baWM · 2026-04-03
> >
> > Most of my questions have been resolved, and the authors can add new discussions and experiments from the rebuttal during the revision process to improve the overall structure of the paper. If it's not easy to transfer the fully open model, I suggest discussing why this method is not easy to transfer to fully open model or what requirements this method has for the basic capabilities of the model (such as grounding or reasoning etc.). This would also facilitate the open-source models in finding directions for improvement.

---

> > > ### Author Response · Authors · 2026-04-07
> > >
> > > Many thanks for your kind reply! After two weeks' work, we are happy to share the results of TVI-CoT on the Bee model with you. We compare TVI-CoT with Bee-8B-SFT upon four MMMU, MMBench, MathVista and MathVerse. Compared to Bee-8B-SFT, TVI-CoT could notably improve the performance on various benchmarks with +2.8% on average. Notably, even Bee has benefitted from high-quality training data on logical problem solving, TVI-CoT still brings + 2.6% and +3.3% performance boost on math benchmarks (MathVista, MathVerse). This shows that TVI-CoT could well generalize to open-source models, and demonstrates that the dynamic token-based revisiting mechanism is the key factor to improve visual reasoning ability. We are also glad to incorporate the contents discussed during the response period into the paper for further revision :).
> > >
> > > | Methods | MMMU | MMBench | MathVista | MathVerse |
> > > | --- | --- | --- | --- | --- |
> > > | Bee-8B-SFT | 66.8 | 83.0| 78.6 | 61.9 |
> > > | TVI-CoT | 69.4 (**+3.6%**) | 84.2 (**+1.2%**) | 81.2 (**+2.6%**) | 65.2 (**+3.3%**) |

---

### Official Review · Reviewer_pDYL · 2026-03-11

**Soundness:** 3
**Presentation:** 3
**Significance:** 3
**Originality:** 3
**Overall Recommendation:** 5
**Confidence:** 2

**Summary:**

This paper proposes TVI-CoT, a novel method to incorporate visual elements in chain-of-though reasoning of VLMs to improve their visual reasoning capabilities. It incorporates learnable control token to allow the model to dynamically look into regions of input image as part of reasoning process. Experiment results show that the proposed method improves performance on multiple visual understanding tasks.

**Compliance With Llm Reviewing Policy:**

Affirmed.

**Final Justification:**

My concerns are resolved. I kept acceptance rating.

**Key Questions For Authors:**

1, What is the performance of MMMU-Pro? This benchmark might better reflect visual reasoning capabilites than MMMU as it is more vision centric and filtered out questions answerable by text-only models.

2. Similary, it is also interesting to see per category breakdown of MathVerse performance improvements. Specifcally, it categorizes the problems into  "Text Dominant,Text Lite,Text Only,Vision Intensive,Vision Dominant, Vision Only" subsets. Further analysis on how the proposed method improve upon each category will strength the impact of this work.

**Limitations:**

Yes

**Strengths And Weaknesses:**

Strength

1. The proposed work proposed an intuitive and interesting paradigm of using interleaved visual reasoning to solve visual understanding tasks, which is novel and can inspire future works.
2. Unlike several other attempts for interleaved visual reasoning, this work adopts the design of selecting existing image patches as opposed to generate new images during reasoning (as explored in several works of unified multi-modal models), which is more efficient and scalable.
3. The proposed method demonstrated strong empirical performance on a wide range of tasks

Weakness

I find no major weakness of this paper

---

> ### Author Rebuttal · Authors · 2026-03-31
>
> 1. Performance on MMMU-Pro
>
> Thanks for your insightful question! We are happy to provide results on MMMU-Pro as follows. During the rebuttal phase, we reproduce the results on multiple backbones including Qwen2.5-VL-7B, Qwen3-VL-8B and Qwen3-VL-32B, and compare them with TVI-CoT to fully demonstrate its effectiveness. Due to memory constraint, we are only able to train Qwen3-VL-32B with LoRA on 8*80G GPUs. As shown in the table, TVI-CoT could notably improve performance of various backbones, with +6.5%, +6.0% and +3.1% performance boost for Qwen2.5-VL-7B, Qwen3-VL-8B and Qwen3-VL-32B. This shows that TVI-CoT could well generalize across different backbones, and substantially improve visual reasoning ability on advanced benchmarks like MMMU-Pro.
>
> | Methods | Qwen2.5-VL-7B | TVI-CoT (Qwen2.5-VL-7B) | Qwen3-VL-8B | TVI-CoT (Qwen3-VL-8B) | Qwen3-VL-32B | TVI-CoT (Qwen3-VL-32B) |
> | --- | --- | --- | --- | --- | --- | --- |
> | MMMU-Pro | 38.1 | 44.6 (**+6.5%**) | 55.6 | 61.6 (**+6.0%**)| 65.3 | 68.4 (**+3.1%**) |
>
> 2. Category breakdown of MathVerse performance improvements.
>
> Thanks for your interesting question! We are willing to share the results on Category breakdown of MathVerse benchmark. We evaluate TVI-CoT and Qwen3-VL-8B with lmms-eval and list the category breakdown results as follows. From text dominant category to vision only category, Qwen3-VL exhibits an overall descending trend on performance, demonstrating its weak ability to excavate beneficial information from visual-dominant inputs. In contrast, compared to Qwen3-VL, TVI-CoT brings considerable performance boost across all categories, demonstrating its effectiveness on visual reasoning. Especially, TVI-CoT achieves more performance boost on visual-biased categories (+5.2% on average for Vision Intensive, Vision Dominant and Vision Only) that text-biased categories (+2.6% on average for Text Dominant, Text Lite and Text Only). This shows that TVI-CoT can more effectively capture key information from visual inputs, wherein the adaptive revisiting mechanism plays a crucial role to incorporate visual cues within CoT to provide better visual evidence.
>
> | Category | Text Dominant | Text Lite | Text Only | Vision Intensive | Vision Dominant | Vision Only |
> | --- | --- | --- | --- | --- | --- | --- |
> | Qwen3-VL-8B | 66.5 | 58.7 | 61.2 | 52.9 | 50.4 | 46.2 |
> | TVI-CoT | 68.7 (**+2.2%**) | 60.4 (**+1.7%**) | 65.0 (**+3.8%**) | 58.0 (**+5.1%**) | 54.6 (**+4.2%**) | 52.4 (**+6.2%**) |

---

> > ### Author Rebuttal · Reviewer_pDYL · 2026-04-03
> >
> > My concerns are fully resolved

---

### Official Review · Reviewer_gygi · 2026-03-11

**Soundness:** 3
**Presentation:** 3
**Significance:** 2
**Originality:** 2
**Overall Recommendation:** 4
**Confidence:** 4

**Summary:**

This paper introduces a framework called Text-Visual Interleaved Chain-of-Thought (TVI-CoT) to achieve multimodal chain-of-thought by allowing to repeatedly access visual features during the reasoning process when needed. The framework includes different modes of reasoning symbolized by ⟨Think⟩, ⟨Look⟩, ⟨Answer⟩ tags. In thinking stage, MLLM performs standard textual reasoning. The MLLM can generate ⟨Look⟩ tag whenever it needs to retrieve information from the original image. In this “looking” mode, an attention based grounding module retrieves a subset of visual tokens and these retrieved tokens are injected to the end of the context and feeded back to the MMLM. For the experiments, the authors finetunes Qwen3-VL-8B-Instruct model using the proposed approach, and achieve state-of-the-art performances on standard benchmarks such as MMMU, MathVerse and ScienceQA, as compared to tested open-source models.

**Compliance With Llm Reviewing Policy:**

Affirmed.

**Final Justification:**

The authors resolved the issues I raised in my review. I am retaining my Weak Accept score.

**Key Questions For Authors:**

1. Do the authors plan to open source the dataset introduced in the paper?

**Limitations:**

yes

**Strengths And Weaknesses:**

Strengths:
- The model achieves state-of-the0art results among the open source models and even achieves to beat Claude-Opus-4.1on MathVista dataset.
- To train their model, the authors construct a new dataset of 150K training examples with interleaved reasoning and grounding annotations.
- Experiments are performed on 8 different benchmarks.

Weaknesses:
- The paper does not mention unified models and their approach to the visual chain-of-thought problem. For example,
Li et al. Zebra-CoT: A Dataset for Interleaved Vision-Language Reasoninn. ICLR 2026
Shi et al. Intrinsic Visual Chain-of-Thought for Multimodal Mathematical Reasoning. ArXiv Octorber 2025.
- In experiments, they only report results of 7B or 8B models. It would be nice to see how bigger model  such as Qwen3-VL-32B performs on the evaluated benchmarks.
- The hyperparameter k specifying the number of top-k retrieved visual tokens is set to 32. However, the paper does not provide any sensitivity analysis on the effect of this hyperparameter to the overall results.

---

> ### Author Rebuttal · Authors · 2026-03-31
>
> 1. Differences to other unified approaches.
>
> Many thanks for your question! We are happy to clarify the difference between TVI-CoT and others. (1) Zebra-CoT introduces a text-visual interleaving CoT dataset, which covers multiple domains include science, visual reasoning, 3D and visual logics. However, most visual CoT annotations in Zebra-CoT rely on auxiliary imagery including sketches, auxiliary lines, diagrams and visual predictions. This demands labor-intensive manual labeling and requires the model to generate additional auxiliary images during inference. Instead, TVI-CoT can dynamically revisit input images via attention-based token selection, removing the need for extra image generation and auxiliary imagery labelling. Moreover, TVI-CoT can dynamically plan when and how many times to revisit input images, instead of only revisiting only once like previous methods. (2) MathCanvas-Instruct is a specially designed large-scale text-visual interleaving dataset for math reasoning. However, it’s limited to the math domain with specially generated mathematical auxiliary images including sketches, auxiliary lines and diagrams. Though abundant in data scale, it can’t be scalable to general domains outside of math problems. Instead, the adaptive revisiting mechanism of TVI-CoT can reason on general domains and achieve notable performance gains across science, math and common VQA problems.
>
> 2. Performance of bigger models like Qwen3-VL-32B
>
> Thanks for your advice. We implement TVI-CoT based on Qwen3-VL-32B and are willing to report the performance as follows. Due to memory constraint, we are only able to train Qwen3-VL-32B with LoRA on 8*80G GPUs. Compared to Qwen3-VL-32B, TVI-CoT could notably improve the performance on various benchmarks with +1.7% on average. Notably, it brings + 1.8% and +2.0% performance boost on math benchmarks (MathVista, MathVerse). This shows that TVI-CoT could well generalize to larger models, and demonstrates that the dynamic token-based revisiting mechanism is the key factor to improve visual reasoning ability.
>
> | Methods | MMMU | MMBench | MathVista | MathVerse | MMStar |
> | --- | --- | --- | --- | ---| --- |
> | Qwen3-VL-32B | 76.0 | 87.6| 83.8 | 76.8 | 77.7 |
> | TVI-CoT | 78.2 (**+2.2%**) | 88.6 (**+1.0%**) | 85.6 (**+1.8%**) | 78.8 (**+2.0%**) | 79.2 (**+1.5%**) |
>
> 3. Sensitivity analysis of the hyperparameter k denoting top-k retrieved visual tokens.
>
> We are happy to provide ablations for the hyperparameter k which denotes the top-k number for retrieved visual tokens as follows. It’s worth noting that when k equals 0, the model would degenerate into pure textual reasoning. We observe that as k increases from 0, the performance continues increasing at first and reaches a peach when equaling 32. Further increasing k slightly hurts the performance, which may be due to the overly introduced visual redundance. We thus set k as 32 by default.
>
> | k | 0 | 8 | 16 | 32 | 64 |
> | --- | --- | --- | --- |--- |--- |
> | MMMU | 58.9 | 62.1 | 63.2 | **64.3**| 64.1 |
> | MathVerse | 55.2 | 58.4 | 59.8 | **60.2**| 59.2 |
>
> 4. Whether plan to open-source the dataset.
>
> Sure, we will release the training code and the constructed dataset to facilitate the community if the paper is accepted. Hope that could help you!

---

> > ### Author Rebuttal · Reviewer_gygi · 2026-04-03
> >
> > The authors resolved the issues I raised in my review. I am retaining my Weak Accept score. Thank you.

---

### Official Review · Reviewer_gEwG · 2026-03-12

**Soundness:** 2
**Presentation:** 3
**Significance:** 2
**Originality:** 3
**Overall Recommendation:** 3
**Confidence:** 3

**Summary:**

This paper addresses an important limitation of multimodal chain-of-thought reasoning: after the initial visual encoding, many MLLMs continue reasoning mostly in text space without explicitly revisiting image evidence. To address this, the paper proposes TVI-CoT, which interleaves textual reasoning with explicit visual re-grounding through <Think>, <Look>, and <Answer>tokens. The idea is intuitive, and the reported gains on several multimodal benchmarks suggest that repeated visual access can help, especially on visually grounded math and science problems.

**Compliance With Llm Reviewing Policy:**

Affirmed.

**Key Questions For Authors:**

1. What is the clearest difference between TVI-CoT and closely related token-based visual revisiting methods?

2. The training data is constructed using Gemini3-Pro–generated reasoning chains, human filtering, and additional grounding augmentation, resulting in about 150K samples. Could the authors provide more details about this pipeline (e.g., filtering criteria, annotation quality, and grounding noise)? More importantly, can the authors run a control experiment where the same constructed dataset is used to train the backbone model without the TVI-CoT mechanism? This would help clarify how much of the improvement comes from the model design versus the additional training data.

3. Can the authors provide more controlled comparisons on closer backbones, given that TVI-CoT uses Qwen3-VL-8B while some baselines are built on Qwen2.5-VL-7B?

**Limitations:**

yes

**Strengths And Weaknesses:**

**Strengths**

- The paper studies a meaningful and well-motivated problem.

- The proposed framework is simple and easy to follow.

- The empirical trend is generally positive, especially on visually intensive subsets.

**Weaknesses**

- The novelty is not fully convincing relative to closely related work that already introduces intermediate visual/perception tokens or revisits visual evidence during reasoning. For example, Introducing Visual Perception Token into Multimodal Large Language Modelgenerates Region Selection and Vision Re-Encoding tokens for additional perception; Perception Tokens Enhance Visual Reasoning in Multimodal Language Modelsintroduces tokenized intermediate visual representations; and Machine Mental Imagery (Mirage) interleaves decoding with latent visual tokens. Since these ideas are similar to inserting <Look>steps to re-access visual information, the paper should clarify what is fundamentally new in TVI-CoT.

- The ablations do not fully establish that the gains come from the proposed adaptive re-grounding mechanism. Removing interleaving, grounding loss, or replacing grounding with random regions mainly shows that the full system outperforms weakened variants. This does not rule out that the improvements stem from extra supervision, additional structural constraints, or longer reasoning traces rather than better visual retrieval.

- The comparison is somewhat difficult to interpret fairly. TVI-CoT is built from Qwen3-VL-8B-Instruct, while important baselines such as VL-Rethinker-7B and Vision-R1 are based on Qwen2.5-VL-7B-Instruct.

- The data construction pipeline is a potential confounder and is not sufficiently detailed. The training set is built from Gemini3-Pro–generated reasoning chains, human filtering, and additional grounding augmentation using Gemini3-Pro and a pretrained grounding model, yielding about 150Ksamples. Given this substantial additional supervision, it remains unclear how much of the improvement comes from the model design versus the constructed training data.

---

> ### Author Rebuttal · Authors · 2026-03-31
>
> 1. Differences between TVI-CoT and recent methods.
>
> Many thanks for your insightful question! We are glad to clarify how TVI-CoT differs from prior work. First, Re-Encoding [1] introduces Region Selection Tokens and Vision Re-Encoding Tokens. If the former is outputted, it triggers a second model pass by inputting cropped image regions, while the latter invokes an extra encoder (e.g., DINO) to re-encode the full image followed by MLLM inference. Both require additional visual encoding and extra inference, leading to high computational cost. In contrast, TVI-CoT performs attention-based token selection during CoT to revisit inputs, eliminating extra encoding and redundant reasoning for higher efficiency. Second, LLaVA-AURORA [2] inserts perception tokens into CoT but relies on heavy pre-trained models (e.g., instance segmentation) and auxiliary images to distill meaningful features. Similarly, Mirage [3] introduces latent visual tokens by aligning them with auxiliary images through an extra encoder. A key limitation of them is their dependence on auxiliary images, which require extra labeling and heavy encoders. TVI-CoT avoids this by revisiting inputs via attention-based token selection, removing the need for auxiliary data or extra encoders. Moreover, prior methods only allow a single revisit to the image, whereas TVI-CoT dynamically determines the number and timing of revisits, supporting a more flexible design
>
> 2. Effectiveness of the adaptive re-grounding mechanism.
>
> Thanks for your key question! To isolate the effectsof TVI-CoT from extra training data, we retrain Qwen3-VL-8B on the same data as TVI-CoT and evaluate on MMMU, MMBench, MathVista, and MathVerse. We find that training Qwen3-VL-8B on our data improves performance by about +1.3% on average, mainly due to the enhanced training pipeline. Gains are more evident on general visual benchmarks (MMMU, MMBench), while only marginal improvements (<1.0%) are observed on math benchmarks (MathVista, MathVerse). In contrast, TVI-CoT consistently outperforms Qwen3-VL-8B across all benchmarks, with especially larger gains on math tasks, surpassing it by +3.1% on MathVista and +2.3% on MathVerse.
>
> | Methods | MMMU | MMBench | MathVista | MathVerse |
> | --- | --- | --- | --- | ---|
> | Qwen3-VL 8B | 58.2 | 83.4 | 56.4 | 73.2 |
> | Qwen3-VL 8B (re-trained) | 60.6 (+2.4%)| 84.5 (+1.1%) | 57.1 (+0.7%) | 74.1 (+0.9%) |
> | TVI-CoT | **64.3 (+6.1%)** | **84.7 (+1.3%)** | **60.2 (+3.8%)** | **76.6 (+3.2%)** |
>
> 3. Fair comparison with the same backbone
>
> We re-implement TVI-CoT on Qwen2.5-VL-7B and make comparisons across four benchmarks. TVI-CoT outperforms competing methods on 3 out of 4 benchmarks with large margins. Notably, on MMMU and MMBench, it surpasses the second-best method by +3.0% and +10.9%, demonstrating strong visual reasoning ability. On MathVista, TVI-CoT performs slightly worse than VAPO-Thinker. However, methods like VAPO-Thinker leverage specially designed RL strategies that significantly boost performance, whereas TVI-CoT does not use RL training. Despite this, TVI-CoT still achieves superior results on most benchmarks. We will further explore incorporating RL training in the future.
>
> | Methods | MMMU | MMBench | MathVista | MathVerse |
> | --- | --- | --- | --- | ---|
> | Qwen2.5-VL 7B | 57.8 | 83.2 | 68.4 | 49.6 |
> | Vision-R1-7B | - | - | 73.5 | 52.4 |
> | VL-Rethinker-7B | 56.7| - | 74.9 | 54.2 |
> | VAPO-Thinker-7B | 60.2 | - | **75.6** | 53.3 |
> | TVI-CoT | **63.2** (+5.4%) | **84.4** (+1.2%) | 73.4 (+3.8%) | **54.6** (+5.0%) |
>
> 4. Details of the dataset construction pipeline
>
> We are happy to provide more details on our dataset construction pipeline and will include them in the paper. We build our training data from two major visual CoT datasets: Visual-CoT and Zebra-CoT. For Visual-CoT, which includes auxiliary bounding boxes, we use Gemini3-Pro to verify whether visual annotations are necessary, whether existing boxes are relevant, and whether additional annotations are needed. When required, Gemini3-Pro generates new bounding boxes, which are further validated by Qwen3-VL-235B-A22B for logical correctness and spatial precision. After this semi-automatic process, 52k samples are retained with updated CoT chains. For Zebra-CoT, we keep samples from 2D visual reasoning tasks, and for the rest, Gemini3-Pro checks annotation correctness and if additional annotations are needed, yielding 43k refined samples. We further expand the dataset using LLaVA-178k by asking Gemini3-Pro to determine if visual evidence is needed and generate text-visual CoT chains when applicable. These annotations are validated by Qwen3-VL, resulting in 55k samples. Finally, we conduct a quality check where Gemini3-Pro and Qwen3-VL-235B-A22B achieve over 97% agreement, indicating high data annotation quality. Two human experts are recruited to verify 5% samples manually, and an overall acceptance rate of 98.7% is reached. We therefore use this semi-annotated dataset for training.

---

### Decision · Program_Chairs · 2026-04-30

**Decision:**

Accept (regular)

**Comment:**

This paper studies the problem of vision-blind reasoning in multimodal understanding for MLLMs. To address this, the authors propose TVI-CoT, a framework that enables explicit interleaving of textual reasoning and visual feature access through learnable control tokens. This allows the model to adaptively switch between textual reasoning and dynamic visual grounding.

During the review and discussion phase, the authors have addressed the questions and concerns raised, including clearer differentiation from unified and thinking-with-image approaches, as well as additional experiments on larger models and shared backbones. Reviewers agree that, with these improvements, this would be a solid piece of work. I recommend acceptance.